

# Identification of regions with a robust increase of heavy precipitation events

Veronika Ettrichrätz[1,2], Christian Beier[1], Klaus Keuler[1], Katja Trachte[1]

[1]Brandenburg University of Technology (BTU) Cottbus - Senftenberg, Chair of Atmospheric Processes, Burger Chaussee 2, Campus Nord, Haus 4/3, Cottbus 03013 Germany
[2]Leipzig University - Institute for Meteorology, Stephanstr. 3, 04103 Leipzig, Germany

*Correspondence to*: veronika.ettrichraetz@uni-leipzig.de

**Abstract.**

Global climate change is increasingly associated with the increase and/or the intensification of extreme weather such as heat waves, droughts, or heavy precipitation events. However, the characteristics and severity of these changes can vary considerably by region and season. This study focuses on heavy and extreme precipitation events over Europe for the time period 1951 - 2099. The main objective is to identify regions which show a robust and therefore reliable change in such events with ongoing climate change. The study is based on daily precipitation values from 40 regional climate simulations of the EURO-CORDEX ensemble with a spatial resolution of 12 km (EUR-11).

Future changes were investigated using four different metrics, which are sensitive to alterations in the number and intensity of the detected events and consider an accumulated precipitation amount over a selected threshold and two return values for a 10- and a 100-year return period. Differences were detected between the climate scenarios RCP4.5 and RCP8.5, between summer and winter half-year, and between three different methods used to identify and quantify temporal changes from a reference period (1951-1980) to a future climate period (2070-2099). Furthermore, two criteria characterizing the robustness of the changes were used, i.e. a dominant agreement on the sign and a majority of significant changes within the full simulation ensemble. The analysis provided relative and absolute changes of the four metrics being used and the area fractions that exhibited a robust change.

With all methods applied, our study clearly confirms a significant increase in heavy and extreme precipitation in northern, central, and eastern Europe and a robust decrease in heavy, but not very extreme events in the southwest of the Euro-CORDEX domain associated with projected future climate change. For large parts of Southern Europe and the Mediterranean, a tendency towards decreasing intensities (down to -11 mm/year for the accumulated threshold exceedance) become visible but without the evidence of robustness.

Both, the intensity and the area with robust changes in heavy and extreme events prove to be significantly stronger in the RCP8.5 than in the RCP4.5 scenario. For Central Europe, for example, the accumulated threshold exceedance increases from 15 mm/year to 24 mm/year and the 100-year return value from 21 mm/day to 30 mm/day. The related robust area fractions extend from 31% to 99% and from 61 % to 95%, respectively. The relative changes are substantial, even averaged over



larger areas, with values greater than 100% (50%) for the severe events and up to 40% (30%) for the very extreme events in the RCP8.5 (RCP4.5) scenario. Heavy events increase relatively more in winter (for Central Europe +117 % for the accumulated threshold exceedance in RCP 8.5) than in summer (+81 %). The opposite is true for the extreme events with a weaker increase in winter than in summer (e.g. for Central Europe +26 % for the 100-year return value in winter and +43 % in summer).

## 1. Introduction

Heavy rainfall events are often associated with floods, which can cause substantial environmental, economic, and infrastructural damages and can also cost lives (e.g., Vellinga et al., 2001). Two years ago, in June, more than 200 people lost their lives in Germany and Belgium due to heavy and persistent rain (Kreienkamp, 2021).

Heavy and extreme precipitation events are expected to increase due to global warming, which causes a generally higher water capacity and enhanced evapotranspiration in a warmer atmosphere. Allen and Ingram (2002) and Trenberth (1999a) considered that the increase in extreme precipitation is related to the general rise of saturation vapor pressure with temperature in the Clausius-Clapeyron relationship, which provides a scaling rate for the maximum moisture content of about 7% per Kelvin. Observations, e.g., by Westra et al. (2013), Fischer and Knutti (2015), and Berg et al. (2013) and modelling studies, e.g., by Rajczak (2017), Ban et al. (2015), Kendon et al. (2014) and Kharin et al. (2013) confirmed that the increase in 24h extreme precipitation is broadly consistent with this scaling rate.

Several studies showed that extreme precipitation has increased recently. Donat et al. (2013) examined 11000 measuring stations worldwide from 1900 to 2010. They highlighted that there are more areas with significantly increasing trends in extreme precipitation amounts, intensities, and frequencies than areas with decreasing trends. Zeder and Fischer (2020) revealed that extreme precipitation increased at almost two-thirds of the stations in Central Europe from 1901 to 2013, although detecting a long-term climate change signal using observational data is challenging. This is especially true at the local scale, due to a limited number of stations, uneven station distribution, and discontinuities (Donat et al., 2013). Rajczak and Schär (2017) evaluated 100 regional climate model simulations and also concluded that the number of heavy precipitation events is increasing across Europe. They pointed out that particularly in autumn and winter over Northern and Central Europe the extreme events increased by about 20%, while in summer over southern and Central Europe the number of rainy days and mean precipitation decreased, but the number of extreme events increased.

In order to develop adaptation strategies to hazards of heavy precipitation, reliable (i.e. robust) knowledge about affected regions, frequencies, duration, and intensity of these events as well as their behavior under climate change is of great importance. Over recent years, various studies using both, model and observational data have been conducted using different approaches to identify and analyze extreme precipitation such as trend analysis (e.g. Zeder and Fischer, 2020; Madsen et al., 2014) or extreme value statistics. In the latter, mostly two methods are used: i) the block maxima method (BMM), as in





Aalbers et al. (2018) and Rajczak and Schär (2017) and ii) the peak-over-threshold method (POT) as in Palacios-Rodríguez et al. (2020) and Berg et al. (2019). BMM (also called the Annual Maximum Method) determines the annual maximum values over a certain period of time and then fits an extreme value distribution (EVD) to these annual maximum values. The advantages of this method are that the values are statistically independent in time and it is less complex (see Tabari, 2021). However, with a short time period (e.g. 30 years), there are only a few data points (30 points for 30 years) to adjust the EVD

and a loss of information may occur. The peak-over-threshold (POT) method requires a predefined threshold. All values exceeding this threshold are used to fit a generalized Pareto distribution (GPD). The advantage of this method is the larger database for fitting the GPD, which allows for a more accurate fit (Tabari, 2021 and Lang et al., 1999). Since we consider short time periods in our analysis (30 years), we decided to use the POT as one method.

In addition to studies that cover entire Europe (e.g. Zeder and Fischer, 2020; Rajczak and Schär, 2017), there are studies that focus only on individual areas - e.g., Adinolfi et al. (2021) for the Alpine region, Hosseinzadehtalaei et al. (2018) for Belgium, Rulfová et al. (2017) for the Czech Republic, Scoccimarro et al. (2016) for the Euro-Mediterranean region, and Soares et al. (2017) for Portugal. Generally, in Central and Northern Europe extreme events increase throughout the year, while the Mediterranean and the entire Iberian Peninsula experience a decrease in the total amount of precipitation. As

reported by Scoccimarro et al. (2016), Soares et al. (2017), and Rajczak and Schär (2017) the number of heavy events increased, except for small areas of the Iberian Peninsula. However, Rajczak and Schär (2017) and Brogli et al. (2019) pointed to high uncertainty, especially in the Mediterranean region. However, there are still open questions concerning the robustness and reliability of the projected changes across different areas of Europe.

A constraint on the robust identification of extreme precipitation in models is the spatial resolution as shown by Ban et al. (2014, 2021), Kendon et al. (2012), Prein et al. (2013), and Pichelli et al. (2021). Extreme precipitation is often small scale, and thus, its representation highly depends on the resolution of the used data. Model simulations are computational cost-intensive (memory and time), which requires a trade-off between horizontal resolution, the number of simulations, and the domain size according to available resources. For small areas, a higher horizontal resolution can be used and thus, more local

effects like small scale convection can be included, while large areas give a general overview of the spatial distribution of precipitation. During the last 30 years, the horizontal resolution of regional climate simulations has been stepwise improved from 50 km used for the PRUDENCE project (Christensen et al., 2007), to 25 km for ENSEMBLES (van der Linden and Mitchell, 2009) and down to 12 km in the EURO-CORDEX project (Jacob et al. 2014). In recent years, simulations have been performed even at a horizontal resolution of 3km (Adinolfi et al., 2021; Ban et al., 2021; Pichelli et al., 2021; Zeman et

al., 2021; Sørland et al., 2021). Ban et al. (2021) analyzed precipitation simulations at a 3 km horizontal resolution and revealed that high-resolution models generally tend to produce more intense precipitation and a lower frequency of wet hours. Pichelli et al. (2021) demonstrated that high-resolution models refine and amplify projected precipitation patterns





compared to coarser simulations and even affect the trend in intensity over some regions. As the number of high-resolution simulations is still small and only single areas of Europe are covered, these studies are not suitable for robustness analysis.

Despite many different studies, to our knowledge, only few addressed robust changes in extreme precipitation across Europe. Rajczak and Schär (2017) considered regions as robust if 90% of the ensemble members have the same sign and the simulations are within the range of observational uncertainty. In Jacob et al. (2014), changes are robust if 66% of the ensemble members have the same sign and show a significant change according to the Mann-Whitney-Wilcoxon test. All these studies use BMM, which can have high statistical uncertainty associated with extreme value analysis (Tabari, 2021).


The main objectives of our study on heavy and extreme precipitation events are the identification of regions which show a robust change of these events in different climate change scenarios and the quantification of the intensity of these robust changes. 'Robust' in our sense is defined by the majority of the signs of the detected changes and their statistical significance. For this purpose, we used the 40 officially provided EURO-CORDEX simulations with a resolution of 11 km

for the entire European area to identify changes in heavy and extreme precipitation events from past and present-day climate to future scenarios. The analyses aim at three aspects (i) the influence of the method being used to detect a robust change, (ii) the impact of different scenarios and (iii) seasonal modifications. The paper is structured as follows: The data used and methods applied are described in section 2 and 3. The main findings are presented in section 4 and are further discussed and compared in section 5.

**2. Data**

For our analysis of heavy and extreme precipitation events, we used gridded daily precipitation data from a consistent ensemble of 40 regional climate simulations over Europe, the so-called EURO-CORDEX ensemble (Jacob et al., 2014, Vautard et al., 2021). It includes results from five regional climate models (RCMs) driven by transient climate change simulations of eight different global climate models (GCMs) from the Coupled Model Intercomparison Project - Phase 5

(CMIP5) ensemble (Taylor et al., 2012). The simulations include two different climate change scenarios (RCP4.5 and RCP8.5) for the period from 1951 to 2100. All data are available for the same domain (see Figure 1) covering whole Europe, parts of East and North Atlantic, the Mediterranean Sea with parts of North Africa, the Black Sea and littoral states, and parts of Russia on an identical, regular, but rotated grid with a spatial resolution of 0.11° (~12 km). According to Table 1, which summarizes the simulation ensemble with its different combinations of RCMs and GCMs and corresponding scenarios, our

study is based on 26 and 14 simulations for the RCP8.5 and RCP4.5 scenarios, respectively. For the detection of temporal changes, we divide the time series into a reference period (1951-1980) and a future period (2070-2099). Furthermore, changes in daily precipitation values are studied on an annual basis and for summer (April - September) and winter (October - March) half years separately. In addition, we use the E-OBS gridded data set (Cornes et al., 2018, version 23.1e) with a resolution of 0.1°, the ERA5 reanalysis (Hersbach et al., 2020) with a resolution of 0.25°, and the KOSTRA-DWD data



(Malitz and Ertel, 2015 and Junghänel et al., 2017) with a resolution of 8 x 8 km for a qualitative comparison with the EURO-CORDEX simulations. While the E-OBS and ERA5 data consist of daily or hourly precipitation values, from which the required extreme values must first be calculated, the KOSTRA data directly provides the extreme values for different return periods which were calculated from high-resolution gridded precipitation data spatially interpolated by a multivariate regression model from station observations in Germany between 1951 and 2010.

For a more detailed spatial analysis, we focus on the eight Prudence regions (Figure 1) described in Christensen and Christensen (2007), which are the British Isles (BI), Scandinavia (SC), France (FR), Central Europe (ME), Eastern Europe (EA), Alpine region (AL), Iberian Peninsula (IP), and the Mediterranean region (MD). It should be finally mentioned, that in the case of the Prudence regions the calculations are restricted to land points only.

## 3. Methods

Three different methods are used to detect and analyze heavy and extreme daily precipitation events (see Figure. 2), which consider (i) trends in annual threshold exceedance amounts, (ii) differences in annual threshold exceedance amounts between different periods, and (iii) differences in return values between different periods.

A heavy or extreme precipitation event is generally defined as an event in which the daily amount of precipitation exceeds a certain threshold value, which is defined by or corresponds to a high percentile of occurrences. In some studies (e.g., Kendon
et al., 2008, Nissen and Ulbrich, 2017), only the rainy days are included in the calculation of these percentiles, while other studies (e.g. Ban et al., 2015, Rajczak and Schär, 2017) consider all days. The differences between wet-day and whole-day percentiles are often large, as Schär et al. (2016) showed for three typically used indices. They recommended that all-day percentiles should be used to assess potential changes in heavy rainfall events. They concluded that in the case of using wet days only, misleading results may occur.


In our study, we define a threshold value for daily precipitation that should be exceeded 3 times per year as a long-term average. This corresponds approximately to the 99.2 percentile of all days. This percentile is also used in other studies, e.g. Lazoglou and Anagnostopoulou (2017) and Anagnostopoulou & Tolika (2012) using the 99th percentile and Berg et al. (2018) using exactly the same threshold of three events per year on average.

The threshold values and the derived trends, differences in exceedance amounts, distribution parameters, return values, etc. (see following sections) are first calculated individually for each grid cell and model simulation. Then, the ensemble median of these quantities is determined over all simulations, but only on grid cells, where the detected threshold is at least 1 mm. Finally, the robustness of the calculated parameters indicating a potential climate change signal is checked for each grid-cell ensemble by two criteria. First, the sign of the climate change signal must be the same for more than 2/3 of the simulations.
Second, more than half of these simulations must show a significant change of the considered parameter on a 5 % significance level.



For the comparison of the thresholds calculated from the simulation ensemble with corresponding thresholds from E-OBS and ERA5 data (in section 4.1), and for comparing the return levels with KOSTRA-DWD (in section 4.2) the grids of the reference results are interpolated to the rotated grid of the EURO-CORDEX domain by the nearest neighbor method of the CDO-tool (Schulzweida, 2022).

### 3.1 Method 1 - Trend method

An increase of heavy precipitation events can mean both an increase in their frequency and/or intensity. To include both effects in our analysis, we do not count the exceedance events but accumulate the daily exceedance amount above the detected threshold for each year and season as a baseline index. The threshold values are determined for every simulation and every grid cell individually over the reference period from 1951 to 1980. The annual (or seasonal) exceedance amounts are then calculated for the whole simulation period from 1951 to 2100. In order to identify a temporal development of this index, a linear regression analysis is performed according to Sen (1968). This method determines the linear slope by the median of all pairwise calculated temporal gradients, and is less influenced by outliers than the standard linear regression where the coefficients are estimated by least squares (Sen 1968, Helsel and Hirsch, 2020). The significance of the slope is checked by a Mann-Kendall test. For a better comparison and interpretation, the calculated slope is expressed in relation to the mean annual exceedance amount in the reference period as a percentage change over 120 years, the distance between the reference and future period.

### 3.2 Method 2 - Difference method

The second method uses the same threshold values and annual and seasonal exceedance amounts as method 1. The temporal change is, however, no longer determined by a regression analysis, but by a direct comparison between the 30 values of two 30-year periods, the reference period (1951-1980), and a future period (2070-2099) towards the end of the simulations, which are identical to the periods used in method 3. The annual/seasonal exceedance amounts are summed up for both periods, and the relative difference between the future and the reference period with respect to the total exceedance amount of the reference period provides the rate of change. The significance of this difference is tested by the Wilcoxon-Mann-Whitney rank sum test between the two samples.

### 3.3 Method 3 - Distribution method

While methods 1 and 2 consider the change in heavy precipitation events that are actually realized by the simulations on average 3 times per year, method 3 focuses on the analysis of extreme events that may never have occurred during the entire simulation period. Such rare events can be identified by using extreme value statistics. For this, we use the POT method, described in detail by Coles (2001). First, the threshold value is determined analog to method 1 but for the two periods separately. In order to guarantee temporal statistical independence, all days except the maximum of a rain episode - a period





with consecutive rain days above 0.1 mm - are removed. Then the episode maxima larger than the threshold are selected, and a generalized Pareto distribution (GPD) is fitted to these values. From the GPD distribution parameters (threshold, shape, scale) the 10- and 100-year return values are calculated. Next, the relative difference of the two return values between the future and the past period is determined with respect to the past. The GPD parameters (shape and scale) and the two return values are estimated by the Python package "threshold modeling" written by Lemos et al. (2020). The fit is optimized by the maximum likelihood method.

In order to check the quality of the GPD fits and the significance of the simulated changes, we conduct four Goodness-of-Fit tests based on the Kolmogorov-Smirnov method for each simulation and grid cell. The first two tests check if the threshold exceeding intensity values of the past (respectively future) period are adequately represented by the fitted GPD. If one of the tests fails, the method seems inadequate and is no longer applied to the corresponding grid cells of the considered simulation. The next two tests check if the threshold exceeding intensity values of the past (respectively future) are also adequately represented by the fitted GPD of the future (respectively past) period. Only if both tests fail, a significant change of the return values is assumed. All tests are performed on a 5% significance level.

## 4. Results

### 4.1 Spatial distribution and evaluation of threshold values

Figure 3 (top row) shows the spatial distribution of the threshold median of the larger RCP8.5 ensemble for three analyzed seasons. The values range from 1 mm/day in the very dry regions of northern Africa up to more than 75 mm/day in areas with a generally higher rainfall amount, particularly along the mountains (e.g. Alps, Pyrenees) and the coastlines. The area median of the thresholds (dots in Figure 4) across the eight prudence regions varies between 18 and 38 mm/day on the yearly timescale, with the highest value in the AL region and the lowest in the SC region (see Figure 1) despite the high values along the Norwegian coastal region. Although the spread between the ensemble members (vertical lines in Figure 4) is considerable, the thresholds are clearly separated between some regions. As a consequence, e.g. a value of 19 mm/day may already represent a severe event for EA in winter, but must still be considered as "normal or weak" in most of the other regions.

The difference in the thresholds between summer and winter depends on the region as well. The mountainous and coastal areas show higher values in winter than in summer, whereas the northern, central and eastern parts of the domain tend to have higher thresholds in summer. With respect to the Prudence regions, the differences in the area medians between summer and winter for the RCP8.5 ensemble are negative for BI (-0.3 mm/day), AL (-0.5), IP (-6.8), MD (-4.8) and positive for SC (6.6 mm/day), FR (0.3), ME (4.4), and EA (5.5) as shown in Figure 4. The results are almost identical for the smaller RCP4.5 ensemble. The differences between regions and seasons are the same, only the spread is slightly smaller in the RCP4.5 ensemble than in the RCP8.5 ensemble due to the smaller number of simulations.





However, most of the simulations produce much higher thresholds than the two reference data sets E-OBS and ERA5 as
demonstrated in Figure 3 (middle and bottom row). The values in this figure indicate to which quantile the threshold of the
reference data corresponds within the whole simulation ensemble. The deviations seem to be strongest over eastern land
regions and the Mediterranean, are stronger in winter than in summer, and also stronger for E-OBS than for ERA5 (except
for ME and BI). Dark blue areas indicate that all simulations produce higher thresholds than the reference data. In contrast, a
tendency to lower values can be seen over water (data only for ERA5) in parts of the Atlantic (towards the coast of northwest
Africa) and in the Mediterranean, more in winter than in summer. This discrepancy between the simulated and reference data
becomes even more obvious when comparing the return values with the reference data in the next section.

## 4.2 Spatial distribution and evaluation of return values

The top rows in Figures 5 and 6 show the spatial distributions of the 10 and 100-year return values for the median over the
RCP8.5 ensemble and the reference period 1951-1980. The spatial patterns look very similar to those of the threshold values
(Figure 3). The 10-year and 100-year return values of the ensemble median vary between 17 and 115 mm/day and between
42 and 185 mm/day respectively on the yearly timescale. As with the threshold values, the return values are particularly high
in the mountains, in the Mediterranean region, and along the coasts. For the Prudence regions (see Figure 7), the ensemble
median of the return values on the yearly timescale varies between 40 and 80 mm/day for 10-year return period and 65 and
120 mm/day for 100-year return period. The median values are particularly high in AL, MD, and IP with 81/119 mm/day,
70/112 mm/day, and 62/98 mm/day for 10/100-year return period. Similar to the thresholds, extreme events are lowest in SC
with values of about 43 mm/day for 10-year return period and 64 mm/day for 100-year return period, despite the Norwegian
coast and mountainous region. The other regions (BI, FR, ME, EA) have similarly high return values between 48 and 52
mm/day for 10-year return period and 72 and 81 mm/day for 100-year return period.

Just as with the threshold values, the difference in return values between summer and winter depends on the region. Again,
along the mountains and coastlines, extreme events are stronger during the winter than the summer half-year, and in the
northern, central and eastern European regions, extreme events are stronger in summer than in winter. For most of the
Prudence regions, except IP and MD, the area medians of extreme events are greater in summer than in winter. For the
regions SC, ME, and EA the differences between summer and winter are rather large (16.5/27 mm/day for 10/100 year return
values in SC, 15.4/31.7 mm/day for ME, 17.4/35.8 mm/day for EA) but somewhat smaller for France (8.5/24.3 mm/day).
For the regions AL (3.3/ 11.1 mm/day) and BI (4.2/10.9 mm/day) the extreme events are also slightly higher in summer than
in winter, although their thresholds (Figure 4) are lower in summer than in winter. For IP and MD, the difference between
summer and winter of the return values is reversed with values of -10.4 and -7.2 mm/day for 10-year return period and -11.4
and -3.1 mm/day for 100-year return period and has again the same sign as the threshold differences.

The comparison of the simulated return values with those calculated from daily precipitation values of the E-OBS data for
the reference period (1951-1980) is given in the middle row of Figure 5 (10-year return values) and Figure 6 (100-year return





values). The values indicate, analogous to Figure 3, to which quantile the return values of the reference data correspond within the entire ensemble of simulated return values. Green colors mean that the return values of the reference data are within the interquartile range of the simulation ensemble. This is mainly the case in Central Europe and for larger parts of Italy, Scandinavia, and the British Isles. For most areas, the 10-year return values of the E-OBS data are in the lowest quartile (light blue) or even below all simulated values (dark blue) especially during the winter half year. This mismatch between simulated and observed return values is generally reduced for the 100-year event (Figure 6). Here, the area where the reference values are within the interquartile range or at least in the two lower quartiles of the simulated ensemble is much larger, especially in winter. For only a few small areas (red color in Figure 5 and 6) the reference data show extremes above the full ensemble range.

The comparison looks different if ERA5 reanalysis data are chosen to calculate the two return values (bottom row in Figures 5 and 6). Particularly in winter and over large parts of eastern and southeastern Europe, the extreme values in ERA5 are much higher than the corresponding E-OBS values. Values below the ensemble range occur predominantly over ocean areas. The highest values in ERA5 are still in the uppermost quartile (orange) of the simulations, but only for a few small areas in the model domain. In general, the comparison with ERA5 looks spatially much more homogeneous than with E-OBS. However, for Central Europe the 10 and 100-year events seem to be slightly smaller in ERA5 than in E-OBS as they are mostly located below the interquartile range of the ensemble.

In general, most of the simulations appear to tend to higher return values than those resulting from the two reference data. But due to the strong differences and spatial inhomogeneity in the comparisons, a third evaluation between the higher resolved KOSTRA-DWD data and the simulation ensemble has been carried out. The KOSTRA-DWD data set directly provides the two return values, but only for the area of Germany and the whole year. Their assignment to the quartiles of the RCP8.5 simulation ensemble is given in Figure 8 and shows, that most of the simulated 10-year return values are lower than and partly even completely below (red colored cells in Figure 8, left panel) the reference data. The 100-year KOSTRA-values, however, are mostly located between the 1st and 3rd quartile and only at a very few individual grid points outside the simulated range.

**4.3 Temporal changes of heavy and extreme precipitation events and their robustness**

The three methods used to identify heavy (method 1 and 2) and extreme (method 3) precipitation events result in four relative temporal changes of the underlying indices, which are quantified in Figures 9 and 10 by the relative slope of the annual exceedance amounts, calculated by the trend analysis of method 1 and projected on a 120-year period (1st column), the relative 120-year difference between the future period (2070-2099) and the historical reference period (1951-1980) of the 30 years accumulated exceedance amounts from method 2 (2nd column), and the relative difference between the two periods of the 10-year (3rd column) and the 100-year (4th column) return values. The displayed values represent the median from each simulation ensemble separated by RCP-scenario (upper and lower row in Figure 9 and 10) and season (Figure 9: whole year, Figure 10: summer and winter half-year). Blue grid points indicate an increase of the displayed quantities, and yellow





to red grid points a decrease, while gray-shaded grid points mean that the whole ensemble exhibits no robust change. Figure 11 shows the relative change and Figure 12 the underlying robust area fractions separately for all four quantities (upper row: threshold based indices 1 and 2, lower row: return-value based indices 3 and 4), the eight Prudence regions, and the three seasons. The corresponding absolute changes can be calculated from Table 2. A relative increase of the 10-year return value (index 3) in region ME for the whole year and the RCP8.5 scenario of 30 % would imply with a conversion factor of 4.82

295 mm/10% an absolute increase by 14.5 mm.

The temporal changes show a clear spatial partitioning into three parts (Figures 9 and 10) for all indices, both scenarios and three periods with a dominating increase in the northern and middle parts of the model domain, a decrease in a small southwestern part close to the Iberian Peninsula, and larger areas with indifferent and not robust changes from the middle to

300 the southern domain. The relative changes in RCP8.5 are up to twice as strong as in RCP4.5. For example in Scandinavia and winter the intensity change of index 2 is 78.6% for RCP4.5 and 158.4% for RCP8.5 (Figure 11). The differences are greater in the northern and central parts of the model domain and smaller in the southern parts such as IP, MD, and AL (Figure 11).

Not only the intensity of change but also the area fraction with robust changes is generally greater in RCP8.5 than in RCP4.5

(Figures 9 and 10). The area with robust changes of the 100-year return values looks still very noisy in the RCP4.5 scenario and becomes more clear and homogenous in RCP8.5 and all three periods. For BI, SC, ME, EA, and FR regions, the robust area fraction in RCP8.5 is between 80% and 100% (Figure 12). For RCP4.5 it is highest in the SC region with values already between 60 and 95%. The robust area fraction is lowest for IP and MD, where most of the changes turn out to be not robust.

The total area with robust changes is generally larger in winter than in summer (especially for indices 1 and 2). In particular,

for BI, FR, ME, and EA the area fraction in winter is up to 80% higher than in summer (Figure 12). Intensity differences between seasons are also not uniform but depend on the index. For example, indices 1 and 2 indicate a positive difference between winter and summer in Figure 11 for ME and climate scenario RCP8.5 with 32% and 36%, respectively. Indices 3 and 4, however, show the opposite difference for this case with -6.3% and -16.8%, respectively. Referring to Figure 10, the robust decrease in the southwest of the model domain (mostly over sea) is stronger but affects a smaller area in winter than

in summer.

The general fact that the temporal changes in heavy and extreme precipitation events and the robust areas of these changes are stronger in intensity and larger in extension in the RCP8.5 scenario than in the RCP4.5 scenario is independent of the selected index. But the intensity and the robust area fraction of these changes and their differences between scenarios and seasons vary with the selected index or method. The robust area fraction seems generally larger for the two return values

(lower row in Figure 12) than for the two threshold-based indices (upper row in Figures 12), especially in the weaker scenario. Furthermore, it is larger for climate changes of the 10-year return values than for the 100-year return values. The strength of the changes also varies with the method (Figure 11). The two threshold-based methods provide the strongest relative increases, while the relative increase of the 10-year return values is still weaker than that of the 100-year values.





The trends of the two threshold-based indices (upper row in Figure 11) show a generally stronger relative increase in winter than in summer for all regions. This is not the case for the two return values (lower row in Figure 11). The 10-year return value confirms the threshold differences between winter and summer trends only for the regions IP and MD, while all other regions show an opposite behavior with stronger (or nearly the same as for AL) relative changes in summer than in winter. The 100-year return values, however, indicate a stronger relative increase in summer or all regions.

## 5. Discussion and Conclusions

The main objective of this study was the identification of robust future climate changes in heavy and extreme precipitation events with three different methods. We analyzed the area fractions with robust changes in selected subdomains (the eight Prudence regions), the intensity of these changes for different scenarios and seasons, and the influence of the methods used. We applied our analysis to daily precipitation values of the largest and most consistent regional climate simulation ensemble (EURO-CORDEX) currently available. The determination of areas with robust changes in the full ensemble was performed here for the first time, and thus represents the unique feature of our study.

The definition of the robustness of a climate change signal is not uniformly fixed and is therefore to some extent arbitrary. Definitions of robust changes in an ensemble of climate simulations are generally based on two criteria, a dominating majority of the sign of the simulated changes and the statistical significance of these changes. In our case, the trend of climate change must be the same for more than 2/3 of the simulations and for more than half of these simulations the detected changes in the considered index must be statistically significant at a significance level of 5%. This is similar to the definitions used in Jacob et al. (2014), where areas were also considered as robust if 2/3 of all simulations show the same trend and the trends are significant. However, the required proportion of significant changes remained unclear in their study. Rajczak and Schär (2017) used a stronger definition of robustness, requiring 90% of all simulations to have the same sign, but they did not apply any statistical significance test to the analyzed changes. Compared to these two studies, we used other and additional methods and indices to determine the climate change in heavy and extreme precipitation, a reference period further in the past, another seasonal differentiation, and a different focus on robust areas.

Four indices were introduced to check to what extent the identification of robust changes was independent of the method used. Furthermore, the indices were selected in such a way that two indices each quantify the changes in heavy and in extreme daily precipitation. We chose the largest possible time interval of 120 years, with the reference period 1951-1980 in which climate change had not yet shown a substantial increase. Intense precipitation can also be triggered by convective events during spring and early fall. To make a clearer distinction between convective and non-convective months, we conducted our analyses for the two half-year periods April to September (referred to as summer) and October to March (referred to as winter) rather than for the usual three-month seasons. Another advantage of the semiannual periods is the larger sample size for the statistical analysis.



The spatial distribution of the threshold values (Figure 3) and the return values (Figures 5 and 6) show predominantly the
        same spatial structure, although they represent clearly different precipitation intensities. Threshold and return values are
        particularly high in areas where generally more precipitation falls, such as mountains due to topographically induced lifting
        or coasts due to coastal convergence. The lower thresholds and return values in continental areas can be explained by lee
        effects and increasing continentality. The seasonal differences in thresholds and return values depend on the region. The

increase of thresholds in Central and Eastern Europe during summer can be explained by more convective events with more
        small-scale precipitation than in winter. The higher thresholds in winter along the mountains and coasts originate from
        larger-scale synoptic systems like fronts which hit Central Europe more often during winter than in summer because large-
        scale advection and atmospheric dynamics are more pronounced in winter than in summer (Trenberth, 1999b; Trenberth et
        al., 2003b; Mikolaskova, 2009). A small seasonal discrepancy occurred between return values and thresholds for BI and AL.

For these two regions, the return values are slightly higher in summer than in winter, while the opposite is the case for the
        threshold values. This could mean that extreme precipitation in these regions is triggered more by convective events, which
        occur predominantly in summer, as mentioned by Ye et al. (2017), while the somewhat weaker heavy precipitation events
        are related more to intense low-pressure systems, which contribute most to precipitation episodes in winter. It should be
        noted, however, that both the resolution and the convection parameterization used by the models may hinder an adequate

reproduction of convective precipitation events, as discussed by Giorgi et al. (2019). Furthermore, simulations over regions
        with highly structured topography such as the Alps are generally problematic at relatively coarse resolution, and can still
        pose problems at higher resolutions, as described by Caldas-Alvarez et al. (2022).

        The comparison of the thresholds (Figure 3) and the return values (Figures 5, 6 and 8) with different reference data provided
        an inconsistent picture. It seems that the simulations generally produced much higher extreme values. But the reference data

itself also showed very large deviations. Compared to ERA5 and E-OBS, the thresholds of the ensemble median were 10-
        20% larger. However, the return values of KOSTRA over Germany were on average 10-20 % larger than those of the
        simulated events calculated with method 3. Surprisingly, the simulated 100-year return values agreed much better with the
        reference data than the 10-year ones. This makes the reliability of the reference data generally questionable. The differences
        to E-OBS can be mostly explained by the variable density of measurement stations and the smoothing interpolation of the

station data to a different regular grid (see Cornes et al., 2018 and Vautard et al., 2021). The largest deviation from E-OBS
        occurred in the eastern part of the model domain. Here, the station density used for the E-OBS data is particularly low, so
        that the large-scale interpolation of daily values by flat cubic splines (Cornes et al., 2018) should lead to a general smoothing
        effect of extreme daily events. In regions with high station density like BI, ME, and SC the deviations were much smaller.
        Deviations to ERA5 were less pronounced and more homogeneous than to the E-OBS data, as the representativeness of the

reanalysis is also more homogeneous for the entire Europe. But the resolution of ERA5 is about 3 times coarser (Hersbach,
        2020) than that of the simulation ensemble and the precipitation values result from short range forecast simulations on that
        spatial scale. The coarser resolution of the simulations compared to KOSTRA could be one reason why the calculated





extreme events of method 3 (for Germany) were smaller than the values of the KOSTRA data (Figure 8). However, this holds only for the 10-year return values. The better agreement between the 100-year values could be a random effect of the greater uncertainty of these values. In principle, the discrepancies between KOSTRA and the simulated return values may be influenced by differences in the analysis period, the methods used to determine the return values, and the remapping of the reference data to the EURO-CORDEX grid. Overall, the difference between the three reference data and the ensemble generates a large uncertainty, so that it remains unclear how reliably these data represent the actual extreme values.

The detection of robustness depends on the choice of index, season, scenario, and region. The reasons for the lack of robustness vary by method. Threshold-based methods failed mainly at significance and return value-based methods failed at 2/3-majority of sign. In regions like IP, MD, and North Africa the thresholds were partly too low, which excluded many grid cells from our analysis. But regardless of the method, season, or scenario, our analysis of robust climate change showed a clear spatial tripartition (Figure 9 and 10) with a dominating intensification of heavy and extreme precipitation in northern to central and eastern parts of continental Europe, a broad zone of non-robust changes in Southern Europe and the Mediterranean to North Africa, and a small area of a substantial decrease in intense precipitation events over the southwest of the Spanish Peninsula and offshore Atlantic. The changes in the sub-domains SC, EA and ME turned out to be particularly robust and large. Precipitation in mid-and high latitudes increases due to, among other things, a poleward extension of the Hadley cell and shifting storm tracks (see IPCC2013, Giorgi et al., 2019). And as Berg et al. (2013) and Ye et al. (2017) had already pointed out, convective precipitation will also increase in these regions. Of particular interest are the regions MD, IP, and North Africa. IP and MD had a much lower proportion of robust cells than the northern areas. These areas are not only affected by a substantial decrease in mean annual precipitation, as mentioned by Rajczak and Schär (2017) and Brogli et al. (2019), but also by a decrease in extreme events. We see this decrease for North Africa as well - but it does not meet our robustness criteria. For IP and MD, however, a decrease in heavy precipitation occurred only in summer (Figure 11). The annual values of exceedance amounts (index 2) for the Mediterranean and Alpine regions increased by about 71% and 86%, respectively (RCP8.5), which is in line with the results of Bonanno et al. 2018, who used a lower spatial simulation resolution to calculate the change and fewer simulations. The 20-30 % increase of return values in the Alps in the RCP8.5 scenario is consistent with the values in Rajczak and Schär (2017), although they used a different calculation method. In the case of the Alps, these calculations are generally problematic for simulations with coarse resolution, since the topography and the related precipitation forcing cannot be adequately resolved and reproduced. In our case, all four indices showed an increase of heavy and extreme events but robust only over a small to medium portion of the Alpine region. The study by Ehmele et al. (2020) revealed a downward trend of exceedance amounts in this region over the period 1900 to 2030. However, looking at the values between 1950 and 2030, their study also showed a clear increasing trend, which is consistent with our changes in the RCP4.5 scenario.



The higher greenhouse-gas radiative forcing in the RCP8.5 scenario leads to stronger warming than in the RCP4.5 scenario in most regions of the world and also in Europe. Rising temperatures in a future climate can generally enhance the hydrological cycle and favor the generation of convective-induced precipitation (see Berg et al., 2013, Ye et al., 2017). Both effects can cause an intensification or accumulation of heavy rainfall events. The results of our analysis confirm these findings for large parts of northern and Central Europe. The RCP8.5 ensemble produced a larger area with robust enhancements of heavy and extreme precipitation events and a stronger relative increase in the intensity of these events than the RCP4.5 ensemble in these parts of the model domain, as Figures 9 and 10 and the analysis of the subregions in Figures 11 and 12 illustrate. However, the southern and Mediterranean parts of the model domain tended more to a decrease in extreme precipitation events, which turned out to be robust only for a small area in the southwest of the model domain. But also this robust area increased and showed a stronger reduction of intense precipitation in the RCP8.5 scenario compared to the 4.5 scenario, particularly in the summer half-year. This happened in accordance with a general decrease in total precipitation in this region for the whole ensemble (not shown here) and may be caused, as explained by Brogli et al. (2019), by changes in the circulation pattern and the atmospheric state mainly in winter, and by an intensification of the land-ocean temperature contrast, a lower relative humidity over land and a reduced moist adiabatic lapse-rate which lead to a higher static stability in summer.

The threshold-based methods showed relative increases of 50% for RCP4.5 and up to 100% or more for RCP8.5. The relative increase was smaller for the return values, but these values generally correspond to higher intensities than the thresholds. An increase in indices 1 and 2 by 100% and more means that the annual amount of precipitation above a threshold, which was exceeded three times per year on average in the historical period, will at least double in the RCP8.5 scenario towards the end of this century. The two threshold-based methods provided the strongest relative increases, while the relative increase for the 10-year return values was still somewhat weaker than that for the 100-year values. The slightly higher relative change for the 100-year return period means that the absolute intensity increase for extreme events is much stronger at 100 than at 10 years since the reference value is already significantly larger.

Regardless of the method and scenario, the robust area fraction is larger in winter than in summer (Figure 10). As already mentioned above, heavy and extreme precipitation in winter is mainly caused by large-scale synoptic events. These events are quite well represented by RCMs with a horizontal resolution of about 12 km. A general increase in climate change should therefore be detected with a high statistical significance and robustness. Extreme events in summer tend to be rather small-scale and of convective nature, especially in Central and Southern Europe. The resolution of the models may be too coarse and the relevant physical mechanisms are not fully captured by the convection parameterization used by the regional climate models on that spatial scale. Hence, the models principally have difficulties representing these events correctly (Berg et al, 2013, Rajczak and Schär, 2017, and Giorgi et al., 2019) and detecting robust changes in a changing regional climate during summer. Ban et al. (2021) showed for example that 3 km high-resolution models tend to have higher precipitation



indices than less high-resolution models. With regard to the relative intensity change in the robust areas, the situation is twofold. While the intensity of heavy events represented by the threshold-based indices 1 and 2 showed a stronger increase in the winter than in the summer half-year, the intensity of extreme and very extreme events represented by the 10- and 100-year return values of indices 3 and 4 increases more in summer than in winter (see Figure 11). For regions IP and MD this is partly in contrast to the results of Rajczak and Schär (2017) who analyzed a decrease in the return values over the three-month summer season. But these deviating results may occur due to different subdivisions of the seasons, another reference period, but most likely due to different calculation methods. The general tendency in the Mediterranean area is in fact a decrease in intensity of the 10- and 100-year return values. But these tendencies turned out in our analysis not to be robust over large parts of the domains. The remaining robust areas were located more to the north and were associated with an increase, resulting on average in a positive tendency also for the extreme return values in these parts.

In summary, despite remaining uncertainties and differences between methods, seasons, and scenarios, the simulation ensemble analyzed by this study projects a substantial increase in heavy and extreme precipitation for large parts of Europe during the 21$^{st}$ century as a result of global climate change. In particular, the regions of northern, central, and eastern Europe, represented by the Prudence domains SC, ME, and EA, are affected by a robust increase over more than 90 % of their areas in the RCP8.5 scenario and over 40 % in the RCP4.5 scenario. However, the biggest uncertainty which remains is the representativeness of the threshold and return-value-based indices in the EURO-CORDEX ensemble in comparison to available reference data. The large discrepancies across Eastern Europe between simulated and observationally derived indices may raise doubts about the reliability of the simulation results. But the homogeneity of the numerous simulations and the larger deviations even between different reference data in this region support the reliability of the regional climate simulations. Over Central Europe, the simulations agree quite well with different reference data and should therefore provide the most reliable results here. Accordingly, the absolute intensity would increase in the RCP8.5 scenario for the accumulated amount above the reference threshold of methods 1 and 2 by about 14 and 24 mm per year on average, and by 14 and 30 mm/day for the 10- and 100-year return values. The more moderate increase in RCP4.5 would amount to only 9 and 15 mm per year for the threshold exceedances and to 9 and 21 mm/day for the return values in Central Europe.

Due to the potential problem that small-scale convection-induced heavy precipitation events in mid-latitudes cannot be adequately resolved by a 12 km numerical grid, it would be interesting to investigate whether the robustness of extreme precipitation changes - especially in summer – can be improved by an ensemble of convection-permitting regional climate simulations with a higher horizontal resolution in the km-scale. The same methodology could also be used to examine the robustness of intensity changes for both shorter (sub-daily) and multi-day precipitation episodes. In addition, the comparison with reference data could be extended to representative station data and the causes for the sometimes considerable deviations in extreme values between reference data and simulation results should be investigated in more detail.





**Table 1: The EURO-CORDEX simulation ensemble used for this study resulting from combination of eight Global Climate Models (GCMs, left column) with five Regional Climate Models (RCMs, first row) for the two greenhouse gas scenarios RCP4.5 and RCP8.5. For further details of the regional and global models and simulations, see Jacob et al. 2014 and Vautard et al. 2021.**

| RCM<br>GCM | CCLM | HIRHAM | RACMO | REMO | WRF |
|---|---|---|---|---|---|
| CanESM2 | RCP8.5 | | | RCP8.5 | |
| CNRM-CM5 | RCP4.5, 8.5 | RCP4.5, 8.5 | RCP4.5, 8.5 | RCP8.5 | RCP8.5 |
| EC-Earth | RCP4.5, 8.5 | | RCP4.5, 8.5 | RCP4.5, 8.5 | RCP8.5 |
| HadGEM2-ES | RCP4.5, 8.5 | | RCP4.5, 8.5 | RCP4.5, 8.5 | RCP8.5 |
| IPSL-CM5A-MR | | | RCP8.5 | RCP8.5 | RCP4.5, 8.5 |
| MIROC5 | RCP8.5 | | | RCP8.5 | |
| MPI-ESM-LR | RCP4.5, 8.5 | | RCP8.5 | RCP4.5, 8.5 | |
| NorESM1-M | | RCP4.5, 8.5 | RCP8.5 | RCP4.5, 8.5 | |

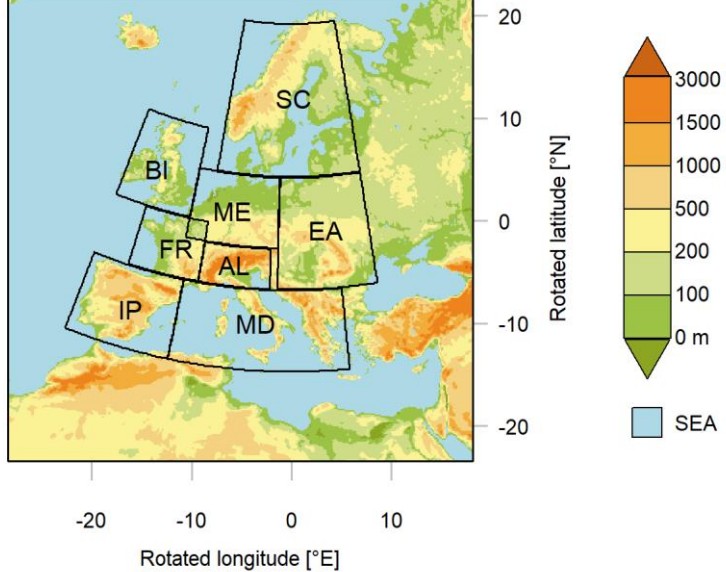

**Figure 1: EURO-CORDEX data domain with a representative orography and the eight Prudence regions: British Isles (BI), Scandinavia (SC), France (FR), Central Europe (ME), Eastern Europe (EA), Alpine region (AL), Iberian Peninsula (IP), and the Mediterranean region (MD), labeled.**





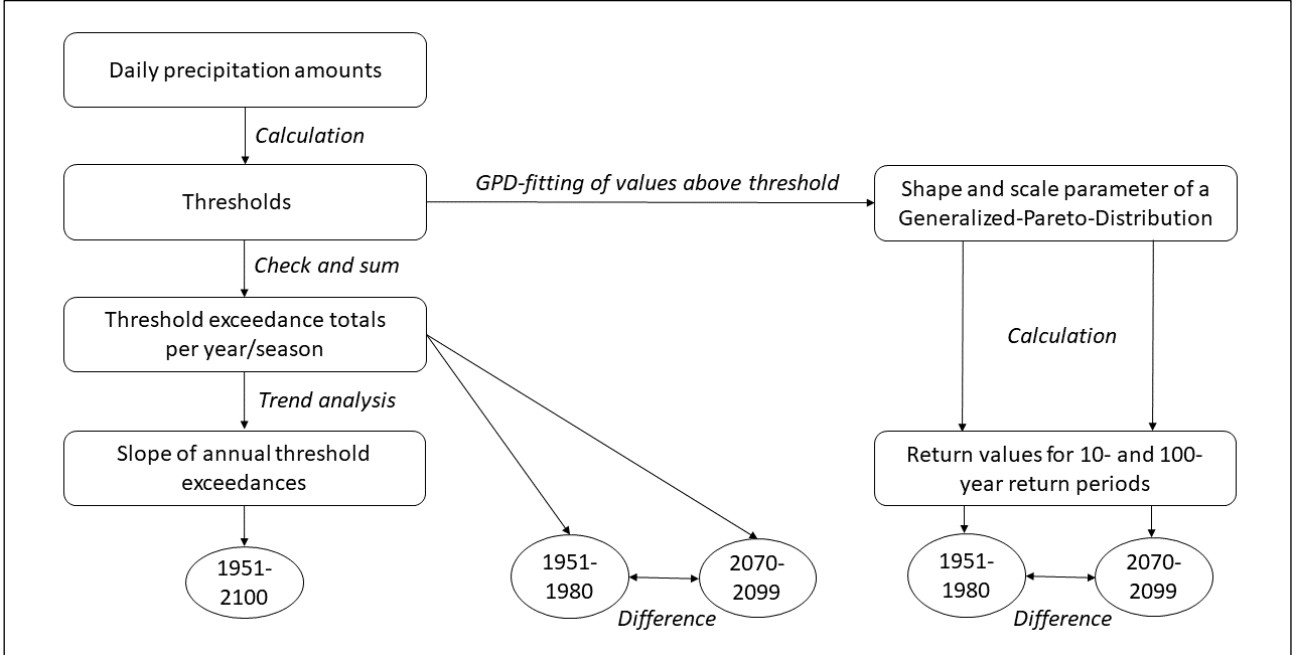

**Figure 2: Methodology flowchart**



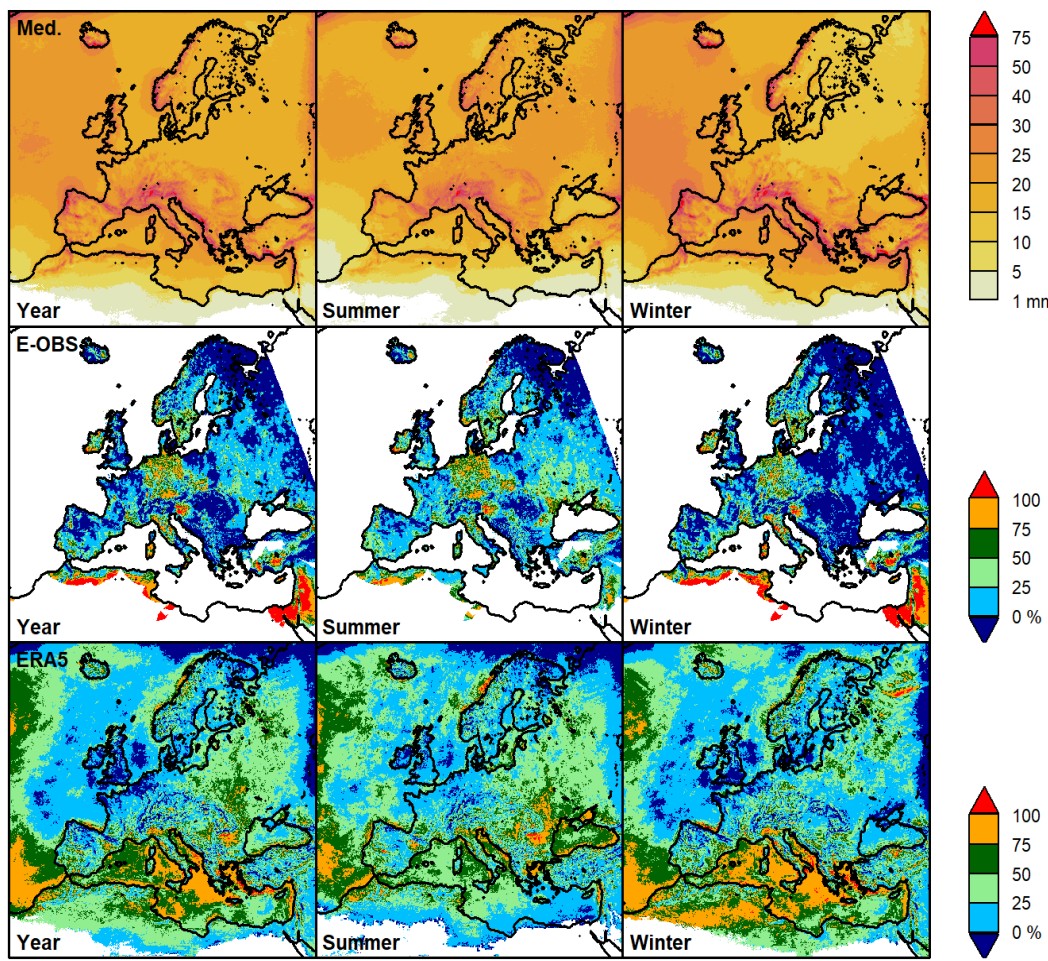

**Figure 3: Spatial distribution of the median of thresholds from the RCP8.5 simulation ensemble (top) for the reference period (1951-1980) and the p-values for E-OBS (middle) and ERA5 (bottom), indicating to which quantile the thresholds from the reference data correspond within the simulation ensemble. White color indicates missing values (E-OBS) or thresholds below 1 mm.**



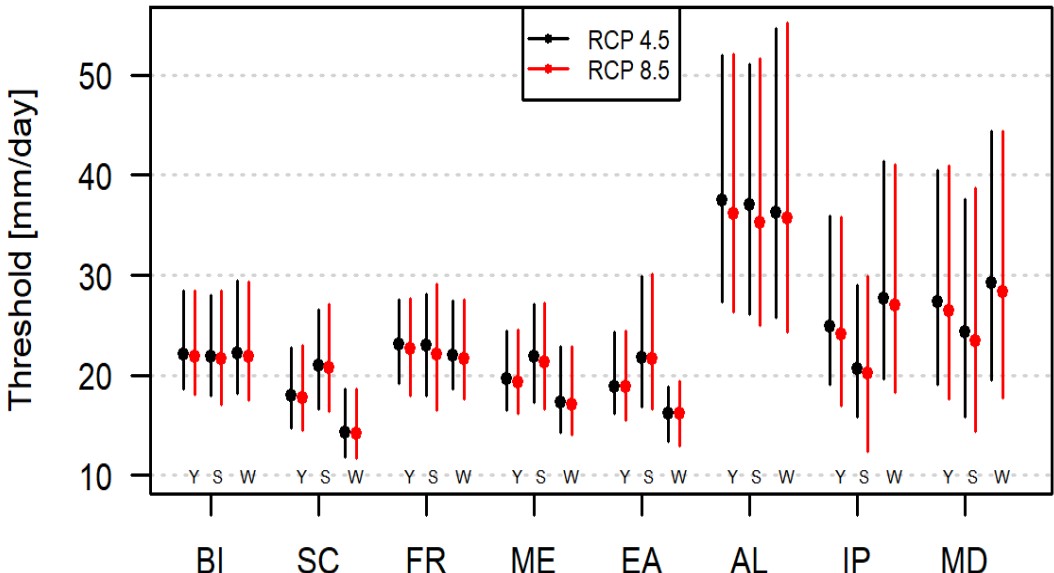

**Figure 4:** Ensemble median (dots) and range from the ensemble minimum to the maximum value (vertical lines) of the simulated thresholds for the reference period (1951-1980). The values represent the area median of these quantities over all grid points of each PRUDENCE region (see Figure 1) for year (Y), summer (S), and winter (W) and for both scenario ensembles (RCP4.5 in black and RCP8.5 in red) in mm/day.



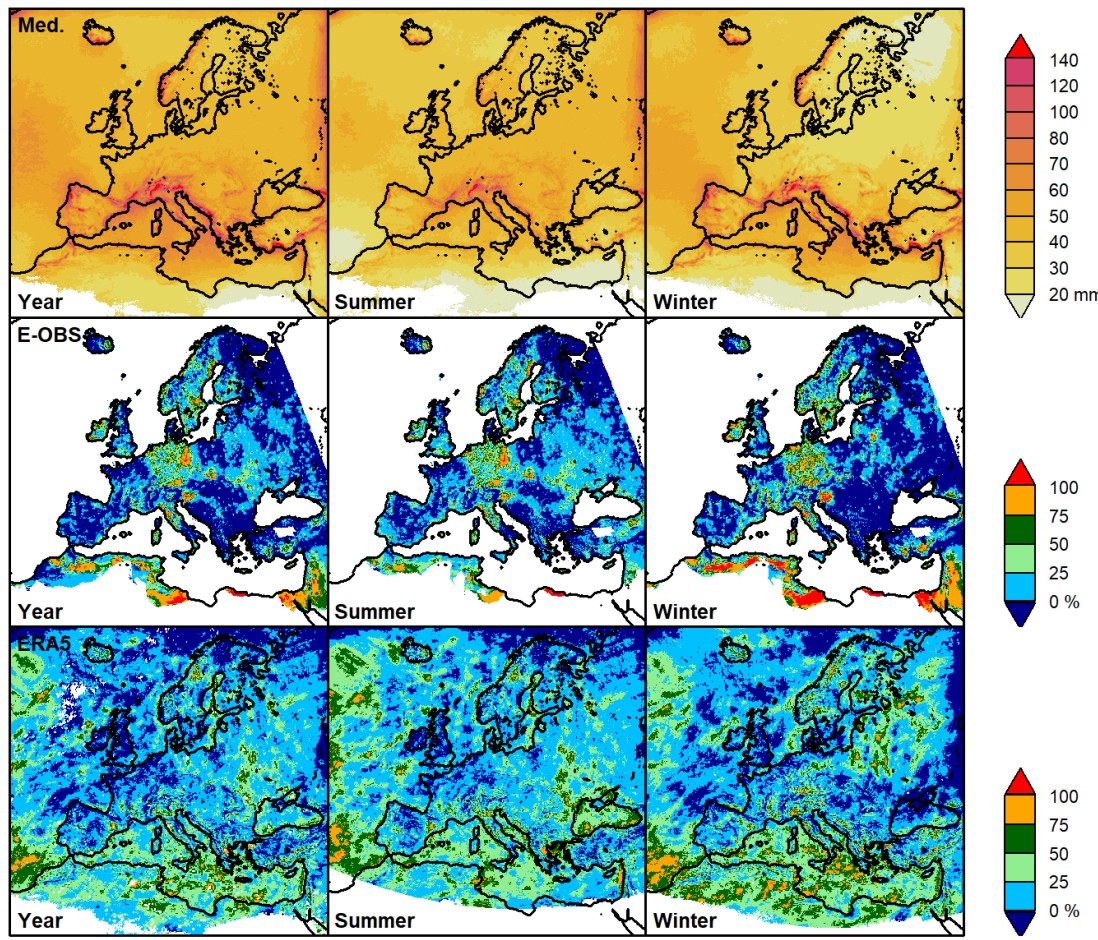

**Figure 5: Spatial distribution of the median of return values for a 10-year return period from the RCP8.5 simulation ensemble (top) for the reference period (1951-1980) and the p-values for E-OBS (middle) and ERA5 (bottom), indicating to which quantile the return value of the reference data corresponds within the simulation ensemble. White color indicates missing values (E-OBS) or thresholds below 1 mm or failing goodness of fit test.**



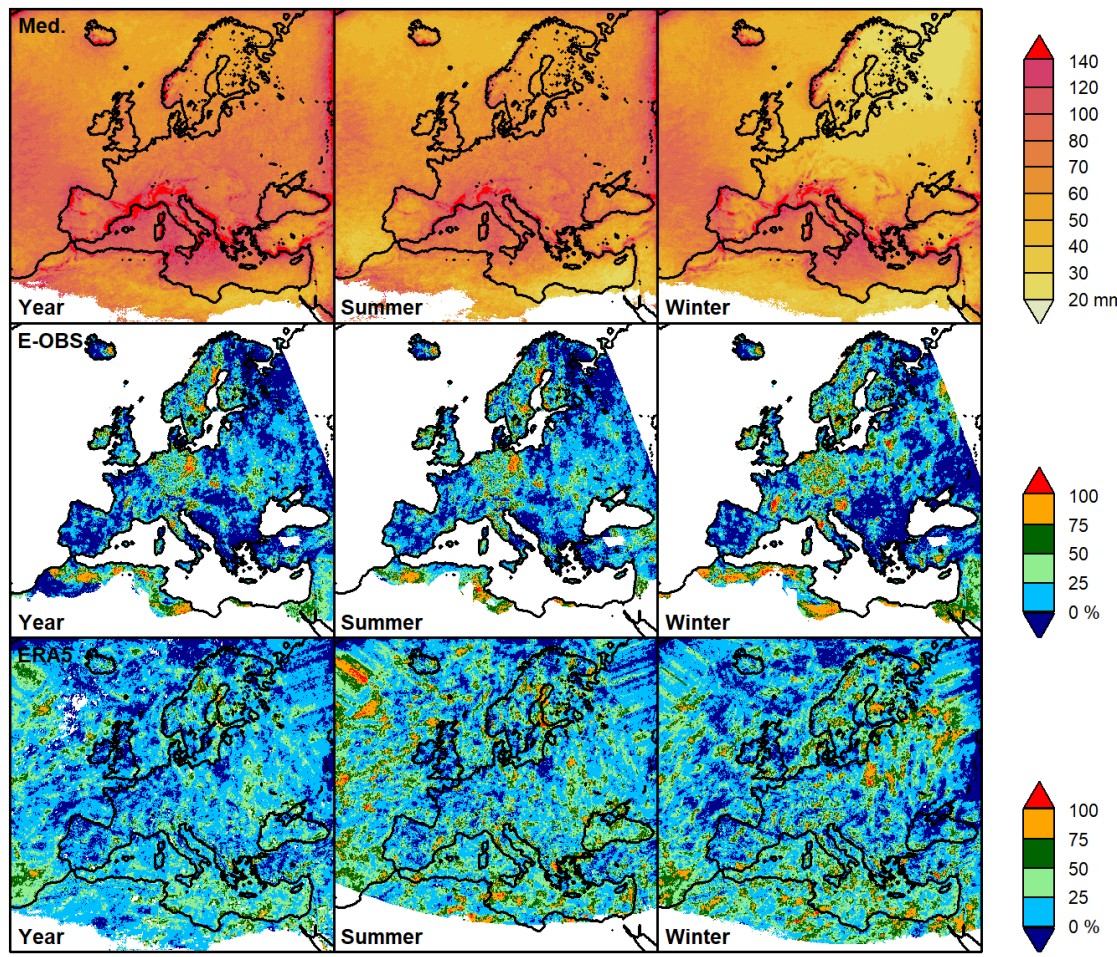

**Figure 6: Same as in Figure 5 but for the return values of a 100-year return period.**





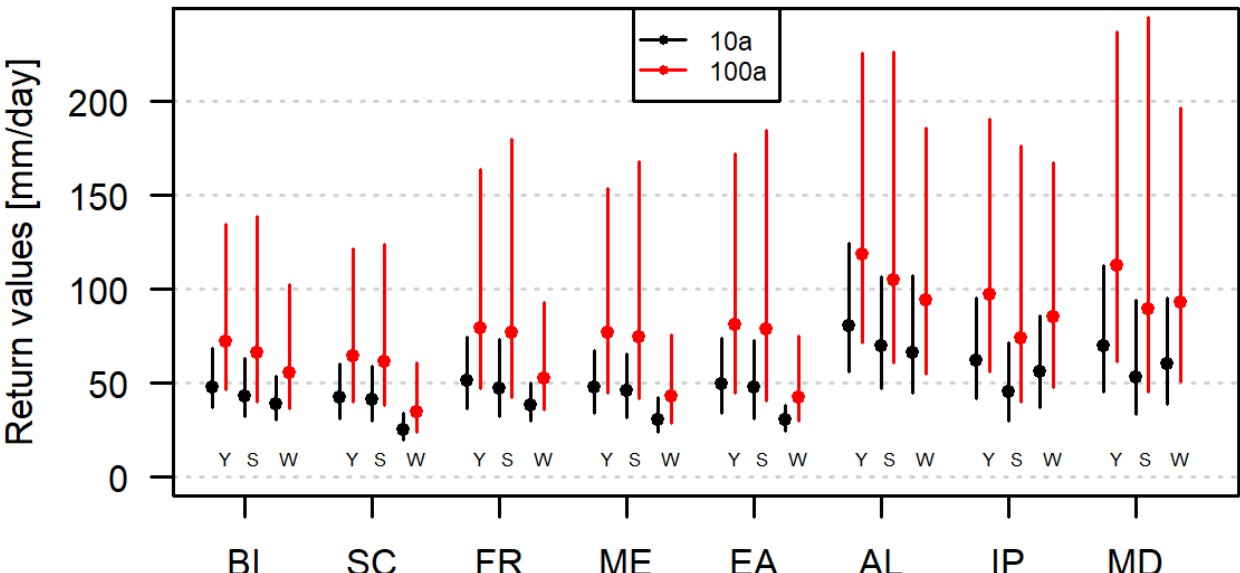

**Figure 7: Ensemble median (dots) and range from the ensemble minimum to the maximum (vertical lines) of the simulated 10-year (black) and 100-year (red) return values for the reference period (1951-1980). The values represent the area median of these quantities over all grid points of each PRUDENCE region (see Figure 1) for year (Y), summer (S), and winter (W) and the RCP8.5 ensemble in mm/day.**



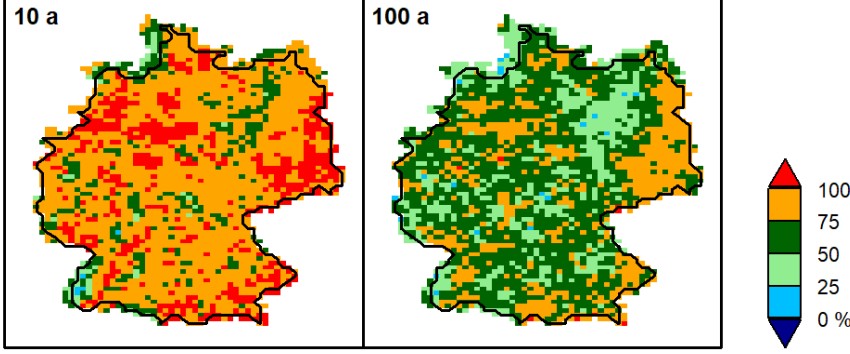

**Figure 8: p-values for the KOSTRA-DWD reference data, indicating to which quantile the reference value corresponds within the simulated ensemble of return levels for 10-year (left) and 100-year (right) return periods. White areas indicate missing values outside of Germany.**





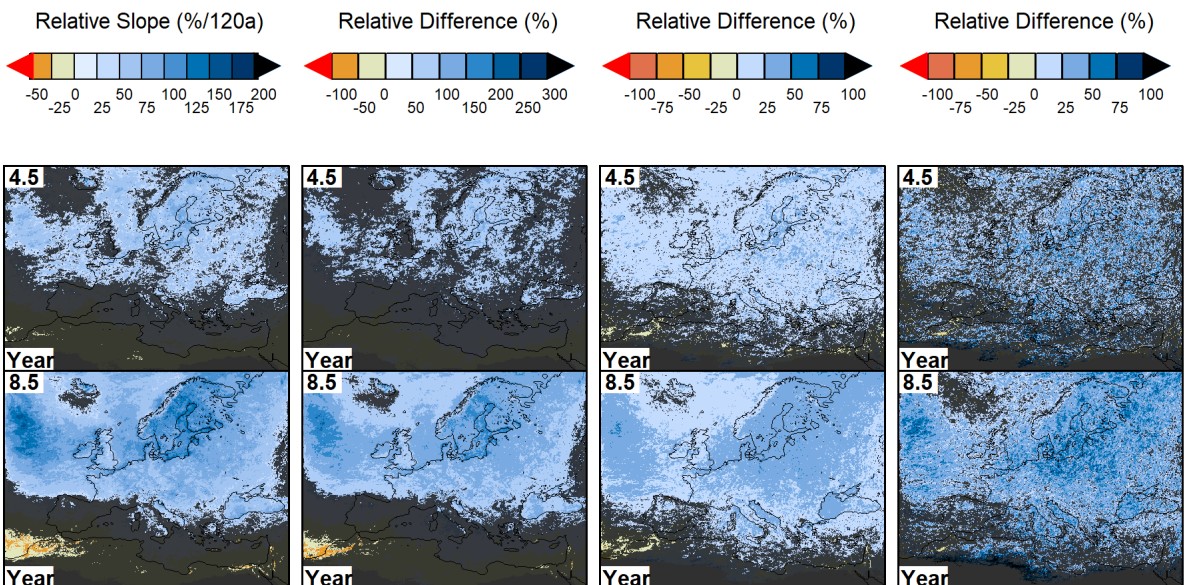

**Figure 9: Climate change of heavy and extreme precipitation events, identified by temporal changes of 4 different quantities calculated for the whole year: the relative slope from the trend analysis of the annual exceedance amounts (method 1) scaled to a 120-year period (1st column) and the relative differences between future (2070-2099) and historic (1951-1980) periods of the accumulated annual exceedance amounts (method 2, 2nd column) and of the 10-year (3rd column) and 100-year (4th column) return values (method 3). Blue areas indicate an increase, yellow to red colors a decrease. Grey shaded areas mark regions where the robustness criteria are not met. The upper row represents the median values of the calculated quantities for the RCP4.5 simulations, the lower row the corresponding values for the RCP8.5 simulations.**





**Figure 10: Same as Figure 9 but for summer and winter period**




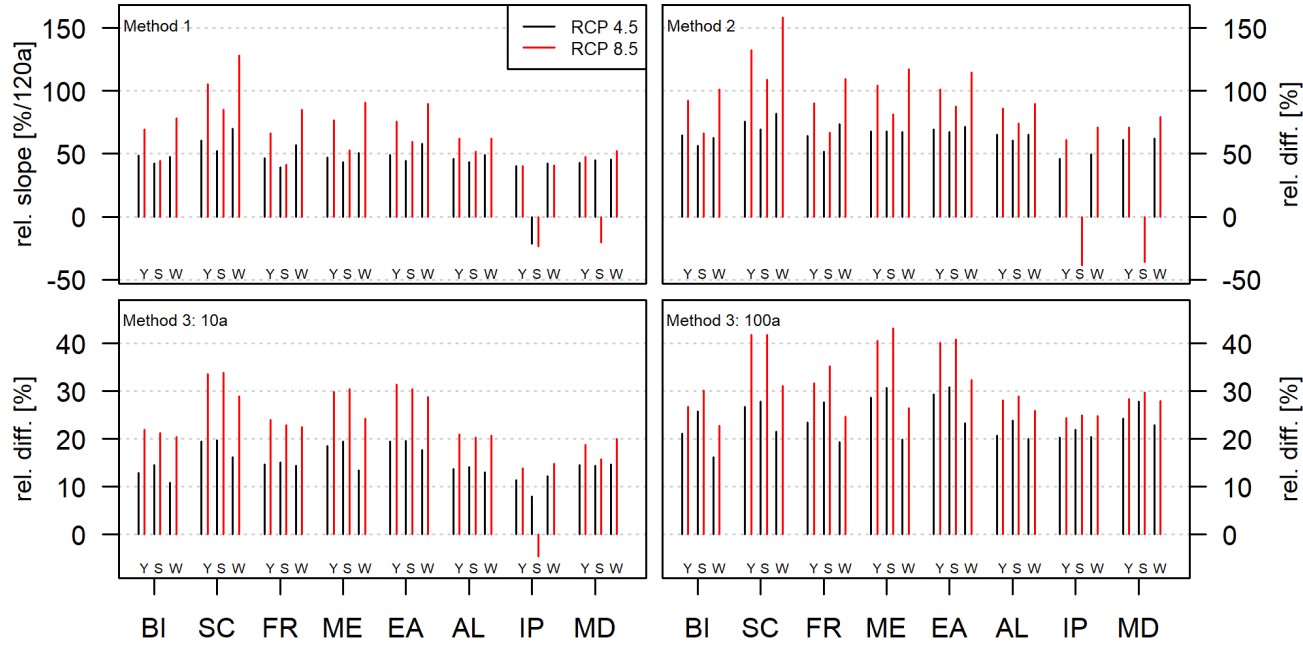

**Figure 11: Area medians of the relative slope in %/120a (method 1, top left) and the relative difference in % of the accumulated threshold exceedances (method 2, top right) and of the 10- and 100-year return values (method3, bottom row) for the medians of the two scenario ensembles (RCP4.5 in black, RCP8.5 in red), the eight Prudence regions and for whole year and the summer and winter half-years. Missing black bars in upper right panel for regions IP and MD in summer indicate that no robust changes were identified for these cases.**




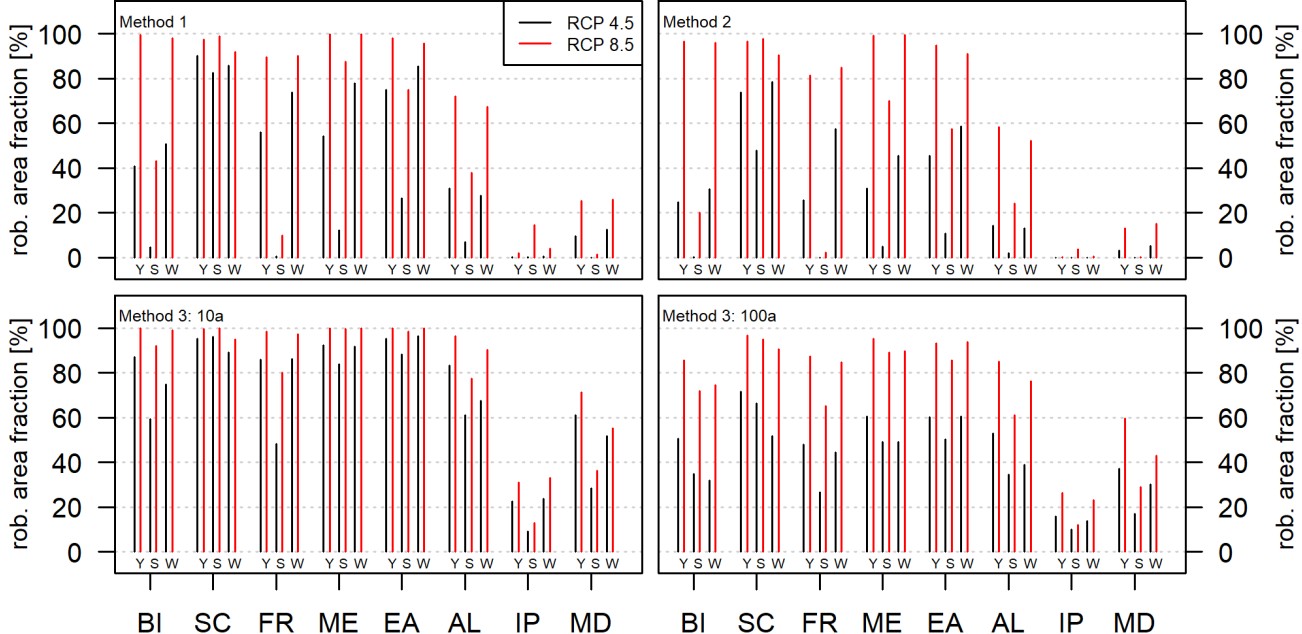

**Figure 12: Ensemble median of area fractions in % with robust changes of method 1 (top left), method 2 (top right) and method 3 for 10- and 100-year return values (bottom row) for the medians of the two scenario ensembles (RCP4.5 in black, RCP8.5 in red), the eight Prudence regions and for whole year and the summer and winter half-years**






**Table 2: Conversion factors to calculate absolute changes in mm for a 10% relative change of the ensemble median for the eight Prudence regions, the four indices, both scenarios, and the three seasons. Missing values at -\* indicate that no robust changes were identified for these cases.**

| Season | Scenario | Index | BI | SC | FR | ME | EA | AL | IP | MD |
|--------|----------|-------|------|------|------|------|------|-------|------|-------|
| year | RCP4.5 | 1 | 2.11 | 2.00 | 2.40 | 2.27 | 2.48 | 3.30 | 2.96 | 3.01 |
| | | 2 | 2.02 | 1.98 | 2.31 | 2.23 | 2.40 | 3.06 | 2.98 | 2.97 |
| | | 3: 10a | 4.96 | 4.26 | 5.34 | 4.86 | 5.08 | 8.10 | 6.23 | 6.92 |
| | | 3: 100a | 7.08 | 6.16 | 8.09 | 7.46 | 7.69 | 11.45 | 9.06 | 10.99 |
| | RCP8.5 | 1 | 2.25 | 1.96 | 2.39 | 2.24 | 2.42 | 3.56 | 2.35 | 2.89 |
| | | 2 | 2.20 | 1.97 | 2.38 | 2.25 | 2.41 | 3.38 | 2.27 | 2.69 |
| | | 3: 10a | 4.86 | 4.20 | 5.17 | 4.82 | 4.97 | 7.90 | 6.04 | 6.65 |
| | | 3: 100a | 6.93 | 6.17 | 7.69 | 7.36 | 7.71 | 11.25 | 8.61 | 10.04 |
| summer | RCP4.5 | 1 | 1.99 | 2.10 | 2.31 | 2.29 | 2.57 | 3.00 | 2.63 | 2.51 |
| | | 2 | 1.81 | 2.04 | 1.74 | 2.19 | 2.44 | 3.03 | -\* | -\* |
| | | 3: 10a | 4.25 | 4.19 | 4.97 | 4.67 | 4.87 | 6.88 | 4.63 | 5.35 |
| | | 3: 100a | 6.32 | 5.88 | 7.58 | 7.27 | 7.41 | 10.00 | 7.35 | 8.75 |
| | RCP8.5 | 1 | 2.10 | 2.06 | 2.31 | 2.35 | 2.51 | 2.99 | 2.18 | 2.50 |
| | | 2 | 2.01 | 2.07 | 2.33 | 2.30 | 2.45 | 2.92 | 2.26 | 2.95 |
| | | 3: 10a | 4.25 | 4.07 | 4.80 | 4.59 | 4.79 | 6.47 | 4.68 | 5.19 |
| | | 3: 100a | 6.19 | 5.84 | 7.25 | 7.04 | 7.30 | 9.62 | 6.68 | 7.85 |
| winter | RCP4.5 | 1 | 1.83 | 1.16 | 1.73 | 1.45 | 1.43 | 2.52 | 2.66 | 2.33 |
| | | 2 | 1.82 | 1.13 | 1.69 | 1.47 | 1.41 | 2.28 | 2.68 | 2.09 |
| | | 3: 10a | 4.07 | 2.59 | 3.89 | 3.11 | 3.05 | 6.31 | 5.53 | 5.58 |
| | | 3: 100a | 5.31 | 3.35 | 5.21 | 4.32 | 4.12 | 8.40 | 7.32 | 8.52 |
| | RCP8.5 | 1 | 1.87 | 1.15 | 1.77 | 1.42 | 1.43 | 3.03 | 2.23 | 2.52 |
| | | 2 | 1.87 | 1.14 | 1.73 | 1.42 | 1.42 | 2.73 | 2.11 | 2.26 |
| | | 3: 10a | 4.02 | 2.53 | 3.81 | 3.10 | 3.01 | 6.23 | 5.20 | 5.40 |
| | | 3: 100a | 5.44 | 3.44 | 5.24 | 4.28 | 4.17 | 8.38 | 7.07 | 7.61 |



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

**Code availability**

Evaluation routines are available on request from the authors

**Data availability**

EURO-CORDEX model data are available for instance through the ESGF Node at DKRZ (https://esgf-data.dkrz.de/search/cordex-dkrz/, last access 02/28/2023) under the Creative Commons

Attribution license CC-BY 4.0, E-OBS data from the European Climate Assessment & Dataset (https://www.ecad.eu/, last access 02/28/2023), ERA5 data from the Climate Data Store (https://cds.climate.copernicus.eu/, last access 02/28/2023) and

KOSTRA-DWD from the DWD data archive (https://opendata.dwd.de/climate_environment/CDC/grids_germany/return_periods/precipitation/KOSTRA/, last access 02/28/2023). All Datasets are publicly available.

**Author contribution**

All authors collaborated and contributed to drafting, reviewing and editing the paper.

VE: Writing: Intro, Data, Method 3, Results, Programming Method 3,

CB: Writing Method 1 and 2, creating all figures, Programming of Method 1 and 2



KK: Writing Abstract, Discussion and Conclusion, Coordination

KT: Discussion of results and final revision of text

**Competing interests**

The authors have no competing interests to declare.

**Financial support**

This research has been supported by the German Federal Ministry of Education and Research (BMBF) with a grant (number: 01LP1902E) The article processing charges for this open-access publication were covered by the BTU Cottbus - Senftenberg.

**Disclaimer**

**Special issue statement**

This article is part of the special issue "Past and future European atmospheric extreme events under climate change". It is not associated with a conference.

**Acknowledgements**

This study was conducted as part of the research program "ClimXtreme", which was funded by the German Federal Ministry of Education and Research (BMBF) within the strategy "Research for Sustainability" (FONA).

The authors thank the following institutions for the execution of global and regional climate simulations and the provision of the simulation results: Canadian Centre for Climate Modelling and Analysis of Environment and Climate Change (CCCma),

Centre national de Recherches Météorologique (CNRM), EC-Earth consortium, Met Office UK, Institut Pierre Simon Laplace & IPSL Climate Modelling Centre, Atmosphere and Ocean Research Institute & Japan Agency for Marine-Earth Science and Technology (MIROC), Max Planck Institute (MPI), NorESM Climate modeling Consortium, COSMO-CLM – Climate Limited-area Modelling Community (CLMcom), Danish Meteorological Institute (DMI), Royal Netherlands Meteorological Institute (KNMI), Climate Service Center Germany (GERICS), National Center for Atmospheric Research

(NCAR).

We further thank the European Centre for Medium-Range Weather Forecasts (ECMWF) for providing the ERA5 data, the EU-FP6 project UERRA for providing the E-OBS data set version 23.1, the German Weather Service (DWD) for providing KOSTRA data and the German Climate Computing Center (DKRZ, Hamburg) for computing and storage resources. We thank in particular Marco Oesting from University of Stuttgart for explaining and discussing with us the statistical tests

and methods.