# Peer review of "Identification of regions with a robust increase of heavy precipitation events"

_EGUsphere, 2023_

## Author Comment (AC1)

Authors reply to Referee 1:
Thank you very much for your comments.

1. Referee comment and authors answer

Referee1: "The topic investigated in this study is pertinent and crucial given the rise in extreme events caused by climate change. However, as the impacts of extreme precipitation over Europe using EURO-CORDEX RCMs have been examined in multiple previous studies, a new study in this field must address a critical knowledge gap to justify publication. While some of the previous studies were cited, others, such as a recent study by Ritzhaupt and Maraun (2022) identifying robust and conflicting projections of mean and extreme precipitation across Europe using different ensembles of climate models (ENSEMBLES and EURO-CORDEX RCMs, HighresMIP, and CMIP3, CMIP5, and CMIP6 GCMs), were not mentioned. Another study by Dosio and Fischer (2017) investigated the robustness of changes in extreme precipitation in Europe using EURO-CORDEX RCMs under different global warming levels. Given the existing knowledge in the literature, the objective of this study may be too narrow to justify publication in NHESS."

Authors: We sincerely appreciate your efforts in pointing out the two articles and highlighting the fact that we did not emphasize our unique points sufficiently.
By incorporating these citations into our introduction, comparison, and discussion, we can enhance the strength of our article. We believe that our study makes significant contributions by addressing the following critical knowledge gaps and modifying certain aspects:
a) Ensuring consistency in simulations: A key aspect that distinguishes our research is the rigorous commitment to maintaining consistency in our simulations. We diligently avoid using a mixture of simulations with different configurations and quality levels. By adhering to a consistent set of simulations, we significantly reduce variance, enhance the significance and robustness of our findings, and thereby strengthen the overall reliability of our results. This meticulous approach guarantees the integrity of our study and strengthens the validity of our conclusions.
b) Integration of different methods and metrics: We have employed various methods and metrics in our analysis, and our study presents a combined analysis of their results. This comprehensive approach provides a more thorough understanding of robust changes in heavy and extreme precipitation.
c) Estimating the significance of climate change: Our study quantifies the proportion of land area in individual subregions that will be robustly affected by robust climate change. This assessment helps in determining the significance of potential climate change for these regions, and thus, enhancing our understanding of the impacts.
d) Discussing result uncertainty: We address the potential uncertainty of results in areas where the full ensemble does not adequately represent observed conditions. Instead of solely comparing simulated extremes with reference data based on absolute or relative differences, we demonstrate instances where the reference data lie outside the entire range of the simulation ensemble.
e) Choice of reference period: To exclude the influence of the rising climate change in the 1980s to 1990s, we have chosen the period 1951-1980 as our reference period. This selection enables us to capture the maximum expected increase along the simulation period.

f) Importance of quantifying regional development: A significant challenge in discussing expected future climate changes is reliably quantifying the regional development of extreme precipitation related to global climate change. Our study acknowledges this challenge and emphasizes the necessity of additional studies that consolidate, improve, or validate these expected changes. Such studies help to narrow the range of future climate changes and identify the areas that will be affected by them.

By addressing these knowledge gaps and modifying certain aspects, our research stands out in terms of the consistency of simulations, integrated analysis, estimation of significance, discussion of result uncertainty, choice of reference period, and contribution to the broader discussion of future climate changes. We firmly believe that these distinct features warrant the publication of our work.

What is new or different in Ritzhaupt and Maraun (2023) and Dosio and Fischer (2018) compared to our study?
We would like to note that Ritzhaupt and Maraun's study was published when we finalized our article so that we unfortunately and unintentionally overlooked their publication. Ritzhaupt and Maraun's study differs from ours in several aspects. Firstly, they incorporate a mixture of all available ensembles, including global, regional, and varying resolutions, degrees of development, and time periods for comparison. In contrast, our analysis deliberately focuses exclusively on the most recent regional simulations from the Euro-CORDEX ensemble.
Robustness is generally defined in different ways throughout the literature also in the mentioned two articles by Ritzhaupt and Maraun (2023) and Dosio and Fischer (2018).
In our approach, robustness is determined by combining a strong agreement in sign and a majority of statistically significant changes. This definition is similar but not identical to that of Dosio and Fischer. We give priority to sign agreement and require that at least 50% of the matched simulations exhibit statistically significant changes. Furthermore, we concentrate solely on the highest level of robustness and do not differentiate between "large" and "small" robustness categories. Instead, we analyze fixed time periods that specifically pertain to medium and severe climate changes, without focusing on weaker changes within the global warming range up to +2°C.
We believe that the reference period used in Dosio and Fischer's study (1981-2010) is too late, as it already encompasses a significant portion of observable climate change that began in the late 1980s. Interestingly, Dosio himself acknowledges in the discussion that an earlier reference period could yield different results.

We will extend the discussion in our study by a detailed examination of the method of robustness and a comparative analysis with Dosio and Fischer's findings, with a particular emphasis on contrasting summer and winter changes. We will also incorporate a comparison with the general findings by Ritzhaupt and Maraun (2023).

2. Referee comment and authors answer
Referee: "The authors mentioned that block maxima (BMM) is associated with high statistical uncertainty in extreme value analysis, and therefore used the peak-over-threshold (POT) method in their analysis. This was due to the short time period of their analysis (30 years). However, looking at Fig. 5 of Tabari (2021), it is clear that the difference between

BMM and POT methods for projected changes in extreme precipitation in terms of magnitude and spatial distribution is very small for Europe."
Authors:

Our test calculations in selected regions have shown that the Pareto Return Values are consistently larger than the GEV Return Values, a finding that aligns with Tabari (2021) and his referenced sources. These studies mention that the BMM method is easier to use and implicitly guarantees the statistical independence of the considered extreme values, but also emphasize some shortcomings, especially for short time periods (see citation below). Despite the relatively minor disparities observed between the Peaks-over-Threshold (PoT) and Block Maxima (BMM) methods for Europe, we have chosen to apply the PoT method because of the advantages of the PoT for short time periods (see citation below), despite its difficulties like finding the right threshold and additionally taking care of statistically independent extreme events. Furthermore, we have computed the 100-year return value, and according to Tabari (2021) the discrepancy between PoT and BMM becomes more pronounced with higher return periods (see citation below). However, it is crucial to emphasize that the difference observed for Europe is not significantly substantial.

Citation: Tabari (2021) ["The major advantages of the BM method are its simplicity and the independence of extracted extremes. Its main shortcoming is that it samples only one event per year which may result in a loss of information as other events than the maximum of a year may exceed the annual maxima of other years. It is particularly problematic when the data record is short (i.e., small sample size). Another drawback of the BM method is the inclusion of some lower observations that are still the maximum value in the year (annual maxima in dry years). Annual maximum precipitation depths of coarse time-resolutions can also be potentially underestimated (Hershfield, 1961, Weiss, 1964, Young and McEnroe, 2003, Yoo et al., 2015, Morbidelli et al., 2017, Morbidelli et al., 2018, Morbidelli et al., 2020)."]

Citation: Tabari (2021) ["The POT method supplies a larger sample size for a more precise distribution parameter approximation compared to the BM method which chooses a single maximum event in each year (Lang et al., 1999). It also functions properly in the case of an asymmetric distribution tail (McNeil and Frey, 2000). However, the use of the POT method involves some analytical complexities, including choosing an appropriate threshold level for extreme events and assuring the independence of extremes."]

Citation: Tabari (2021) ["A comparison between global median changes in extreme precipitation intensity with return periods ranging from 2 to 50 years derived from the BM and POT methods for the whole year shows almost same changes up to the return period of 15 years (Fig. 1). For the return periods of longer than 15 years, the ensemble multi-model medians for the BM and POT methods diverge, with slightly larger BM-based changes (up to 2.5% for a 50-year return period)."]

3. Referee comment and authors answer
Referee: "The end of the 21st century (2070-2099) was used for future climate, and the period 1951-1980 was used as the reference period. However, this reference period is not representative of either the current past or the pre-industrial era. The periods 1971-2000 or

1976-2005 are commonly used for climate change studies using EURO-CORDEX RCMs and using these periods facilitates a comparison of the results with previous studies. "

Authors: The reference period of 1951-1980 was deliberately chosen because it precedes the noticeable and significant regional climate changes that occurred in the late 1980s and 1990s. This period corresponds to the decades of economic reconstruction following World War II and represents a time of relatively minimal transient (stationary) climate change. In contrast, the period from 1971 to 1990 encompasses the most noticeable increase in climate change, particularly during the 1980s and 1990s.
This selection provides an advantage over Dosio and Fischer's reference period of 1981-2010, which is already influenced by strong climate changes. By utilizing the reference period of 1951-1980, we align closely with Ritzhaupt and Maraun, who also employed an early period in their analysis of the HighresMIP ensembles.
We will add a justification in our study why we have chosen this reference time.

4. Referee comment and authors answer
Referee: "The authors used 26 and 14 simulations, respectively, for the RCP8.5 and RCP4.5 scenarios. Therefore, the difference in projected changes between the two scenarios may be due in part to their significantly different number of simulations."

Authors: Despite potential variations stemming from the number of simulations conducted, we employed a bootstrap test to determine the minimum number of members required for a grid-point to exhibit robustness under the RCP8.5 scenario. The robust area fraction becomes stable from 10-15 ensemble members depending on region. We also repeated our analysis for the RCP8.5 scenario with a reduced ensemble, where we considered only the same GCM-RCM combinations as used for the RCP4.5 scenario. The relative deviation between the climate change from the reduced and full ensemble is less than 5% for most parts of the model domain apart IP and MD, where the robustness is weak at all.
Our finding aligns with previous studies that have also observed stronger and more robust characteristics associated with the RCP8.5 scenario compared to RCP4.5.
Furthermore, the study by Dosio and Fischer (2018) acknowledge the potential impact of ensemble size on robustness. However, they note that the results for different warming levels, such as 2°C warming computed using only the RCP.8.5 runs, are similar to those obtained using the entire ensemble. This suggests that the influence of ensemble size on the results can be neglected above a certain number of members (about 10-15). Nonetheless, we will explicitly address the issue of different ensemble sizes in our study to ensure transparency in our analysis.

Citation: Dosio, A., & Fischer, E. M. (2018). ["The robustness as defined in our methodology is dependent on the models' ensemble size; the results for the 3°C warming are computed with a smaller ensemble compared to those for 1.5°C and 2°C warmings, which may affect the results. However, from a sensitivity analysis, it turns out that the results, for example., for 2°C warming computed using only the RCP8.5 runs are similar to those using the whole ensemble."]

5. Referee comment and authors answer

Referee: "The authors conducted trend analysis using the Sen slope estimator and tested its significance using the Mann-Kendall test. However, the existence of auto-correlation in time series can influence the results of these methods. For example, positive auto-correlation increases the chance of rejecting the null hypothesis of no trend and vice versa. It is unclear whether the authors checked for auto-correlation in the time series and, in the case of significant auto-correlation, whether its influence was taken into account in trend analysis methods."

Authors:

While Sen's method does not mention any specific issues related to autocorrelation. Helsel and Hirsch (2020) pointed out that ignoring autocorrelation can lead to an overestimation of the accuracy of any statistical estimates.

It is worth noting in this context that a certain degree of autocorrelation is inherent in any trend. A strict linear trend is always characterized by a significant and nearly linearly decreasing autocorrelation. Assessing the presence of autocorrelation in our data, we specifically examined the autocorrelation of annual values in some of the simulations. However, we did not find any significant and structured correlations. The resulting values exhibited the typical characteristics of a strongly noisy linear trend, with autocorrelation values consistently below 0.2 and no lag-dependent structure, except for a region in the northern Atlantic. This particular region displayed the most significant trend overall and exhibited higher autocorrelation values of up to 0.5 for lags between 2 to 4 years, which then decreased afterwards.

We add a corresponding comment on autocorrelation in the revised version.

Citation: Helsel, K., & Hirsch, R. M. (2020). [„One of the consequences of autocorrelation is that the accuracy of any statistical estimates will be overstated if this property is ignored"]

---

## Author Comment (AC2)

Authors response to referee 2:

Thank you very much for your constructive feedback. We appreciate your valuable suggestions, and incorporating them into our work will undoubtedly contribute to the improved quality and comprehensibility of our content.

1. Referee comment and authors answer

Referee:

"There is a clear shift in the quality of argumentation between the chapters 2-4 und chapter 5. In the chapters 2-4 the information given often left me puzzled why the authors choose to do it that way (unusual choice of the reference period, reference data with clear deficiencies for the purpose, partly somewhat arbitrary metrics,…). Finally, in chapter 5 some of it is put into a context. I would suggest to give some of the context earlier at the appropriate places and consider it in the result section. This might change some of the sparsely given explanations in the result section and put the descriptions there in some context earlier on."

Authors:

"In the revised version the chapters will be modified according to your comments and incorporate important explanations in appropriate places beforehand. For example, in Section 2 "Data," we will explain why we have chosen a non-typical time reference and the reference data and in Section 3, where we explain the methodology, we will add an explanation for why we chose this specific metric. By providing this explanation early on, readers will gain a clear understanding of our approach from the outset."

2. Referee comment and authors answer

Referee:

"The four indices given lack a systematic approach and proper discussion. The authors distinguish between "heavy" and "extreme" precipitation. However, there seems to be no systematic approach to that, for instance covering the range of return periods. Furthermore, for "heavy" precipitation, they use a) linear trends and b) the difference between two time slices. The manuscript does not explore thoroughly how the differences in methodology affect the results. Is the trend always linear, for all ensemble members? Does the trend depend on the climate sensitivity of the GCM/RCM chain? Is there an effect of long-term climate variability? A regional shift in the sign of the change over time? Going into some of these topics would make the paper more valuable."

Authors:

"Heavy events are generated directly by the simulations in any case, occur on average 3 times per year or half year and can be handled by a direct statistical analysis (counting of occurrences per year). Heavy events represent return periods of 4 respectively 2 months. Extreme events are much rarer and their range is not sufficiently covered by the simulation for a direct statistical analysis over 30-year periods but is statistically estimated by extreme value theory. The idea behind this distinction is to investigate if all types of intensities show a comparable development. The effect of the different methodologies is explained and discussed in chapter 4.3 lines 316-328 and in chapter 5 lines 434-441. In the revised version we will emphasize more how the differences in methodology affect the results.

We did not test the linearity of the trend but we applied with Sen's (1968) test a not strictly linear but monotonic trend analysis. We derived a mean temporal gradient from this analysis

to enable a comparison with the gradient calculated by the difference method over a 120-year interval.

The dependence of the trends on the climate sensitivity is an interesting aspect. We have analyzed if specific GCM/RCM combinations lead to a systematically stronger increase in heavy precipitation events than others, and we will add a corresponding paragraph in the revised version.

To analyze the effect of long-term climate variability and temporal shifts for the whole ensemble is a complex task, as this variability is given by the GCM's individual variability and may differ in phase and period between the models. We have additionally analyzed with method 2 (difference method) for selected GCM/RCM combinations how stable the calculated tendencies are with respect to an increasing time interval. The results show a widely monotonous increase on decadal scales of threshold exceedance amounts with the strongest increase towards the end of the 21st century for GCM/RCM combinations with a high and low climate sensitivity."

3. Referee comment and authors answer

Referee:
"Lines 116ff, Table 1: The EURO-CORDEX ensemble used here is more an ensemble of opportunity than a balanced one (cf. Sobolowski et al., 2021, DOI: 10.5281/zenodo.7673400). There are for instance about twice as much rcp8.5 than rcp4.5 simulation included. In addition, some GCMs and RCMs are more often represented than others. Who does this affect the robustness of the results? Does the lack of consistency in some regions arise from more the GCM or the RCM spread?"

Authors:
"The statement of Sobolwski et al. is correct. The EURO-CORDEX ensemble does not represent a balanced but a very consistent ensemble of regional climate simulations. But we checked the influence of different ensemble sizes on our results.

We employed a bootstrap test to determine the minimum number of members required for a grid-point to exhibit robustness under the RCP8.5 scenario. The robust area fraction becomes stable from 10-15 ensemble members depending on region. We also repeated our analysis for the RCP8.5 scenario with a reduced ensemble, where we considered only the same GCM-RCM combinations as used for the RCP4.5 scenario. The relative deviation between the climate change from the reduced and full ensemble is less than 5% for most parts of the model domain apart IP and MD, where the robustness is weak at all.

Our finding aligns with previous studies that have also observed stronger and more robust characteristics associated with the RCP8.5 scenario compared to RCP4.5.

Furthermore, the study by Dosio and Fischer (2018) acknowledge the potential impact of ensemble size on robustness. However, they note that the results for different warming levels, such as 2°C warming computed using only the RCP.8.5 runs, are similar to those obtained using the entire ensemble. This suggests that the influence of ensemble size on the results can be neglected above a certain number of members (about 10-15). Nonetheless, we will explicitly address the issue of different ensemble sizes in our revised version to ensure more transparency in our analysis."

Citation: Dosio, A., & Fischer, E. M. (2018). ["The robustness as defined in our methodology is dependent on the models' ensemble size; the results for the 3°C warming are computed

with a smaller ensemble compared to those for 1.5°C and 2°C warmings, which may affect the results. However, from a sensitivity analysis, it turns out that the results, for example., for 2°C warming computed using only the RCP8.5 runs are similar to those using the whole ensemble."]

4. Referee comment and authors answer

Referee:

"Lines 127-128: For the Mediterranean extreme precipitation season the summer to winter separation might be disadvantageous, since it often starts in September (e.g. Grazzini et al., 2019). Could shifts in the heavy precipitation period between the ensemble members affect the lack of "robustness" for that area?"

Authors:

"It is possible that the robustness of changes in the Mediterranean could increase in the winter-half year if we shift this period by one month earlier to September. This is supported by the annual analysis, which is dominated by the most extreme events from September to December and which shows a slightly stronger robust area fraction than in winter half-year but only for the MD (not IP) region and the 10- and 100 year extremes (Method 3) (see Figure 12). However, this would imply a region- and model-specific definition of half-year periods."

5. Referee comment and authors answer

Referee:

"Chapter 2/4.1/5.: The problems with the quality of the reference data are discussed in chapter 5. You might consider if you use some of the statements from chapter 5 for your analysis, which might exclude some areas from a comparison at least with E-Obs. Furthermore, how adequate is ERA5 as a reference? Since - as stated in chapter 5 – ERA5 precipitation is a forecast product and does not include assimilated observed precipitation directly. It would be good to explain your choice of references already in chapter 2. Furthermore, in chapter 4.2 the manuscript shows, that the reference data are outside the range of the observations over large areas. These areas partly differ for both reference data sets. E.g. or E-OBS it seems, that for Germany, where the dataset is based on many stations, the result is quite different compared to Eastern Europe, where it is based fewer station data. In addition, the comparison with KOSTRA-DWD for Germany is not that bad. It seems necessary to put these findings into a perspective. Is the ensemble not suitable or are the references not suitable for the specific requirements? Or do you deem it as not crucial?"

Authors:

"The analysis is most reliable for Central Europe. The additional comparison with ERA5 should demonstrate the substantial uncertainty of the reference data in large parts of the domain where observational reference data are based on fewer stations. We think that the references are not suitable for an analysis of extreme amounts over large parts of the domain, particularly if daily amounts from sparse stations are smoothed by flat interpolation algorithms over large areas. The KOSTRA-DWD dataset should have the highest reliability as it is particularly created for extreme events. A fair comparison would require a Europe-wide data set of the same quality, but this is currently not available. We express this more clearly in the conclusion of the revised version."

6. Referee comment and authors answer

Referee:

"Chapter 3.3, Line 199ff: The text is a bit confusing. Consider reformulation. Since the 100-year return values are far outside the range of the 30-year input data, confidence intervals should be given."

Authors:

"The basic idea of extreme value statistics is to derive long-terme return values from shorter time periods with generally much smaller extreme events. Because these estimates have a high uncertainty, we have introduced a double check procedure to guarantee that the calculated GPD is actually represented by the simulated precipitation events of the corresponding time period and if the simulated extreme events of the future period do not fit the GPD of the historical period (and vice versa). We rephrase the paragraph as suggested. The potential confidence interval can be estimated from the range of return values in the full ensemble. We modfied Figures 4 and 7 so that the bars represent the area median of the confidence interval for the median"

7. Referee comment and authors answer

Referee:

"The authors give criteria for the non-applicability of the methods or the exclusion of certain grid points. How does this affect the "robustness" of the results in such areas?"

Authors:

"If methods are not applicable, for example because precipitation amounts are generally too low (see white areas in Fig 3, 5 and 6 over North Africa), or the calculated GPD does not pass the goodness of fit test (contributes to grey areas in Fig. 9 and 10), a grid point can be completely excluded from the analysis (this is the case if the non-applicability applies for more than 33% of the simulations) and the area ratio of robust changes for a particular sub-region is reduced, which means that we can identify robust changes only for a reduced or minor part of the particular region."

8. Referee comment and authors answer

Referee:

"Chapter 4 structure: Chapter 4.1 and 4.2 are mostly about evaluation, whereas chapter 4.3 is about the climate change signals. You could consider separating them."

Authors:

"Chapter 4 represents three different aspects of the analysis of results, and we would like to keep this structure."

9. Referee comment and authors answer

Referee:

Line 40: "the people died due the flooding caused by extreme precipitation."

Authors:

"Corrected as suggested."

10. Referee comment and authors answer

Referee:

Line 95: "Sørland et al. (2021) is not about CPM simulations"

Authors:

"Corrected as suggested."

11. Referee comment and authors answer

Referee:

Line 151ff: "With a threshold of 3 events per 6 month, you consider a 2-monthly return period in method 1 and 2. Consider stating that to get an easier distinction between your terms "heavy" and "extreme" precipitation"

Authors:

"We clarify this in the revised version. See our answer to the 2nd referee comment above."

12. Referee comment and authors answer

Referee:

Line 240: "..extreme events are lowest in SC…" Consider reformulation like e.g. "..least intense.."

Authors:

"Corrected as suggested."

13. Referee comment and authors answer

Referee:

"Line 248: "…median s of extreme events are greater.." should be changed to "higher"

Authors:

"Corrected as suggested."

14. Referee comment and authors answer

Referee:

"Line 316-318: Long somewhat confusing sentence. Consider reformulation."

Authors:

"Corrected as suggested."